# Efficient Multi-Task Reinforcement Learning via Selective Behavior Sharing

## Abstract

The ability to leverage shared behaviors between tasks is critical for sample-efficient multi-task reinforcement learning (MTRL). While prior MTRL methods primarily focus on parameter and data sharing, these methods do not exploit the fact that learning agents often benefit from sharing behaviors when acquiring skills. Few behavior-sharing methods exist but are limited to task families requiring only directly shareable and similar behaviors. Our goal is to extend the efficacy of behavior-sharing to more general task families that could require a mix of shareable and conflicting behaviors. Our key insight is an agent's behavior across tasks can be used for mutually beneficial exploration. To this end, we propose a simple MTRL framework for identifying shareable behaviors over tasks and incorporating them to guide exploration. We empirically demonstrate how behavior sharing improves sample efficiency and final performance on manipulation and navigation MTRL tasks with conflicting behaviors and is even complementary to parameter sharing. Result videos are available at https://sites.google.com/view/qmp-mtrl.

## 1 Introduction

Imagine we are simultaneously learning to solve a diverse set of tasks in the kitchen, such as cooking an egg, washing dishes, and boiling water (see Figure 1). Several behaviors are similar across these tasks: interacting with the same appliances (like the fridge or faucet) and navigating common paths across the kitchen (like going to the countertop). While solving a particular task, humans can easily recognize the behaviors that can or cannot be shared from other tasks. This enables us to *efficiently* solve multiple tasks by mutually beneficial exploration.

Can we replicate how humans naturally learn multiple skills at once, by noticing and utilizing common behaviors between them, and create a framework that can do the same efficiently? While typical works in multi-task reinforcement learning (MTRL) exploit sharing policy parameters (Vithayathil Varghese & Mahmoud, 2020) or relabeled data between tasks (Kaelbling, 1993), behavior-sharing is underexplored and can lead to complementary improvements. Recent works (Teh et al., 2017; Ghosh et al., 2018) learn a shared policy distilled (Rusu et al., 2015) from all tasks, and either use it directly or to enforce learning of similar behaviors. However, these methods share behavior uniformly across tasks, limiting their effectiveness for task families requiring conflicting optimal behaviors from the same state. In this work, our goal is to extend the efficiency gains of behavior sharing to such task families.

Concretely, we propose the problem of *selective behavior sharing* for improving exploration in MTRL. Our key insight is that an agent's past or current behaviors across tasks can be helpful for exploration during training, despite potential conflicts in the final policies, as shown in human learners (Tomov et al., 2021). For instance, while boiling water in Figure 1, a household robot can start by exploring behaviors found rewarding in other tasks, such as going to the countertop and turning the faucet on, instead of randomly exploring the entire kitchen.

---

Submitted to 37th Conference on Neural Information Processing Systems (NeurIPS 2023). Do not distribute.

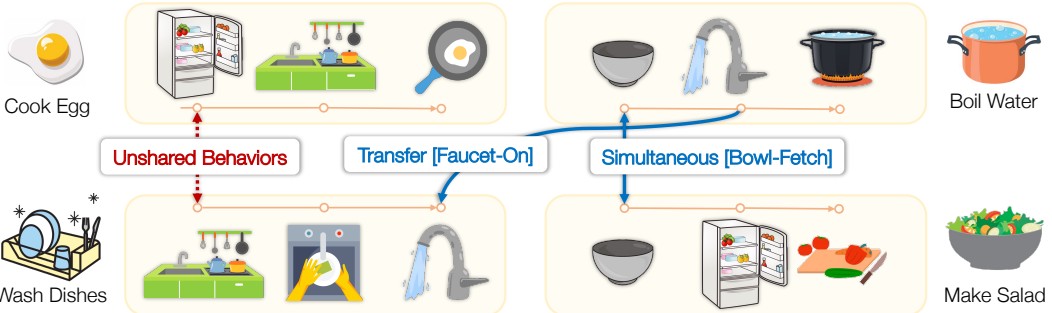

Figure 1: When an agent learns multiple tasks together, selective behavior-sharing can improve overall learning efficiency. (Right) **Simultaneous Learning**: `Boil-Water` and `Make-Salad` tasks can learn the behavior of [`Bowl-Fetch`] simultaneously. (Center) **Transfer Behaviors**: The agent first learns the [`Faucet-On`] behavior in `Boil-Water` task, which can be reused to speed up exploration while learning the `Wash-Dishes` task. (Left) **Unshared Behaviors**: `Cook-Egg` and `Wash-Dishes` require conflicting starting behaviors of going to the refrigerator or sink, not suitable for sharing.

Two key challenges arise in selectively sharing exploratory behaviors for MTRL: identifying and incorporating shareable behaviors. First, the agent must assess the relevance of behaviors from other tasks depending on the current state and training progress. For instance, in Figure 1, other task policies may be disparately helpful (in blue) or harmful (in red) depending on the state of the environment. The second challenge is that initially helpful behaviors can eventually be suboptimal for the task. Thus, the prior approaches that use other tasks' reward-labeled data directly or regularize the policy output to copy other task behaviors would likely fail as tasks diverge. Therefore, we need an effective mechanism to incorporate other task behaviors as exploration proposals.

To address these challenges, we propose a simple MTRL framework called Q-switch Mixture of Policies (QMP), consisting of a Q-switch for identifying shareable behaviors and is used to guide an exploration scheme incorporating a mixture of policies. First, we use the current task's Q-function (Sutton & Barto, 2018), a state and training-progress aware metric, to assess the quality of other task policies' behaviors when applied to the current task. This Q-switch acts as a filter (Nair et al., 2018) to evaluate the potential relevance of explorative behaviors from other tasks. Second, we replace the data collection policy for each task with a mixture of all task policies gated by the Q-switch. Importantly, the mixture is only used for exploration while each policy is still trained independently for its own task. Therefore, QMP makes no shared optimality assumptions over tasks.

Our primary contribution is introducing the problem of selective behavior sharing for improving exploration in multi-task reinforcement learning requiring different optimal behaviors. We demonstrate that our proposed framework, Q-switch Mixture of Policies (QMP), identifies shareable behaviors from other tasks and incorporates them to make exploration efficient. This enables sample-efficient multi-task learning in manipulation and navigation tasks. Finally, we demonstrate how behavior sharing is complementary to parameter sharing, a typical way of improving MTRL.

## 2 Related Work

**Multi-Task Learning for Diverse Task Families.** Multi-task learning in diverse task families is susceptible to *negative transfer* between dissimilar tasks that hinders training. Prior works combat this by measuring task relatedness through validation loss on tasks (Liu et al., 2022) or influence of one task to another (Fifty et al., 2021; Standley et al., 2020) to find task groupings for training. Other works focus on the challenge of multi-objective optimization (Sener & Koltun, 2018; Hessel et al., 2019; Yu et al., 2020; Schaul et al., 2019; Chen et al., 2018), although recent work has questioned the need for specialized methods (Kurin et al., 2022). In a similar light, we posit that prior behavior-sharing approaches for MTRL do not work well for diverse task families where different optimal behaviors are required, and thus propose to share behaviors via exploration.

**Exploration in Multi-Task Reinforcement Learning.** We share the motivation of improving exploration in MTRL with several prior works. Bangaru et al. (2016) proposed to encourage agents to increase their state coverage by providing an exploration bonus. Zhang & Wang (2021) studied sharing information between agents to encourage exploration under tabular MDPs with a regret

guarantee. Kalashnikov et al. (2021b) directly leverage data from policies of other specialized tasks (like grasping a ball) for their general task variant (like grasping an object). In contrast to these approaches, we do not require a pre-defined task similarity measure or exploration bonus. Instead, we learn to identify shareable behaviors and use them for improving exploration in online MTRL.

**Sharing in Multi-Task Reinforcement Learning.** There are multiple, mostly complementary ways to share information in MTRL, including sharing data, sharing parameters or representations, and sharing behaviors. In offline MTRL, prior works selectively share data between tasks (Yu et al., 2021, 2022). Sharing parameters across policies can speed up MTRL by learning shared representations (Xu et al., 2020; D'Eramo et al., 2020; Yang et al., 2020; Sodhani et al., 2021; Misra et al., 2016; Perez et al., 2018; Devin et al., 2017; Vuorio et al., 2019; Rosenbaum et al., 2019) and can be easily combined with other types of information sharing. Most similar to our work, Teh et al. (2017) and Ghosh et al. (2018) share behaviors between multiple policies through policy distillation and regularization. However, unlike our work, they share behavior uniformly between policies and assume that optimal behaviors are shared across tasks in most states.

**Using Q-functions as filters.** Yu et al. (2021) uses Q-functions to filter which data should be shared between tasks in a multi-task setting. In the imitation learning setting, Nair et al. (2018) and Sasaki & Yamashina (2020) use Q-functions to filter out low-quality demonstrations, so they are not used for training. In both cases, the Q-function is used to evaluate some data that can be used for training. Zhang et al. (2022) reuses pre-trained policies to learn a new task, using a Q-function as a filter to choose which pre-trained policies to regularize to as guidance. In contrast to prior works, our method uses a Q-function to evaluate explorative actions from different task policies to gather training data.

# 3   Problem Formulation

Multi-task learning (MTL) aims to improve performance when simultaneously learning multiple related tasks by leveraging shared structures (Zhang & Yang, 2021). Multi-task reinforcement learning (MTRL) addresses sequential decision-making tasks, where an agent learns behaviors or strategies to act optimally in an environment (Kaelbling et al., 1996; Wilson et al., 2007). Therefore, in addition to the typical MTL techniques, MTRL can also share *behaviors* to improve sample efficiency. However, current behavior sharing MTRL approaches (Section 2) assume that the optimal behaviors of different tasks do not conflict with each other. To address this limitation, we seek to develop a behavior-sharing method that can be applied in more general task families for sample-efficient MTRL.

**Multi-Task RL with Behavior Sharing.** We aim to simultaneously learn a multi-task set of $T$ tasks. Each task $T_i$ is a Markov Decision Process (MDP) defined by the tuple $(\mathcal{S}, \mathcal{A}, \mathcal{T}, \mathcal{R}_i, \rho_i, \gamma)$, with shared state space $\mathcal{S}$, action space $\mathcal{S}$, transition probabilities $\mathcal{T}$, and discount factor $\gamma$. The reward function $\mathcal{R}_i$ and initial state distribution $\rho_i$ varies by task. We parameterize the multi-task solution as $T$ policies $\{\pi_1, \pi_2, \cdots, \pi_T\}$, where each policy $\pi_i(a|s)$ represents the action distribution for a given state input and quantifies the agent's behavior on Task $T_i$. The objective of the agent is to maximize the average expected return over all tasks, where tasks are uniformly sampled during training.

Importantly, we do not make the assumption that optimal task behaviors coincide. Optimal behaviors of any two tasks, $\pi_i^*(a|s)$ and $\pi_j^*(a|s)$, can be different at the same state $s$, and thus not directly shareable. Direct behavior-sharing between $T_i$ and $T_j$, such as sharing reward-labeled data (Kalashnikov et al., 2021a) or behavior regularization (Teh et al., 2017), would lead to suboptimal policies.

# 4   Approach

Our approach for behavior sharing is based on the intuition that humans learn to solve tasks by utilizing their knowledge from other tasks (Tomov et al., 2021). To realize this insight in multi-task reinforcement learning agents, we propose to selectively share behavior from other tasks to improve exploration. Two practical challenges arise from this goal:

- **Identifying shareable behaviors.** Behaviors from other task policies should be shared when they are potentially beneficial and avoided when known to be conflicting or irrelevant. Therefore, we need a mechanism to evaluate behavior-sharing between each pair of tasks.

- **Incorporating shareable behaviors.** Having determined the shareable behaviors, we must effectively employ them for better learning. Without a reward relabeler, we cannot share data directly. So, we need a mechanism that can use suggestions from other task policies as a way to explore.

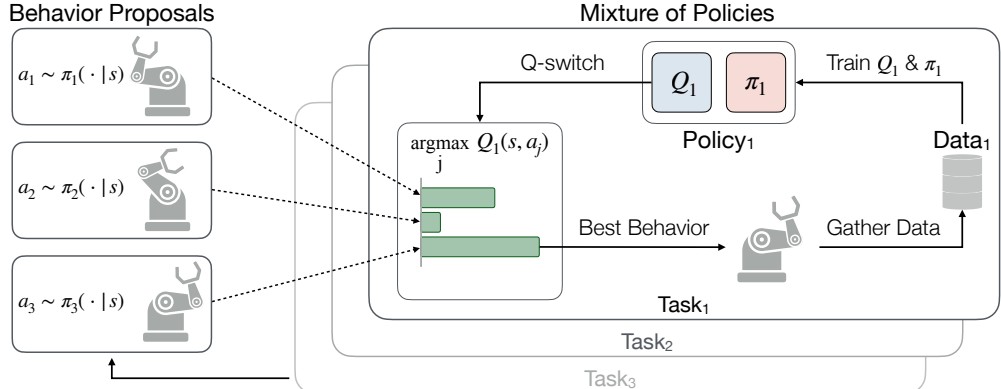

Figure 2: Our method (QMP) trains a policy for each task and facilitates behavior sharing in the data collection phase using a mixture of these policies. For example, in Task 1, each task policy proposes an action $a_j$. The task-specific Q-switch evaluates each $Q_1(s, a_j)$ and selects the best behavior to gather reward-labeled data for training $Q_1$ and $\pi_1$. Thus, Task 1 will be boosted by incorporating only high-reward shareable behaviors into $\pi_1$ and improving $Q_1$ for subsequent Q-switch evaluations.

### 4.1 QMP: Q-switch Mixture of Policies

Our approach to address these challenges is inspired by human multi-task learning. Before trying a new task, we first acquire a general understanding of the effectiveness of different behaviors in achieving the task. We can then identify the most promising behaviors to try and iteratively refine our solution and understanding of the task objective. For instance, when attempting to open a cabinet, we might recall applicable behaviors like approaching the handle or walking away. By comparing their relative effectiveness, we may choose to try the handle. We propose QMP (Figure 2) a novel method that follows this intuition with two components. A Q-switch (Section 4.2) relatively ranks behavior proposals from a mixture of task policies (Section 4.3) which is used as an exploration mechanism.

### 4.2 Identifying Shareable Behaviors

Similar to how a human learning a new task may try out the wrong skill a few times before landing on the correct skill, an RL agent does not know which behaviors are beneficial for sharing between tasks at first. This is simply because it does not yet know the optimal behavior or understand the task objective. So we can only identify shareable behaviors by *estimating* the value of different behaviors based on our current experience and continue to update this estimate as we become more proficient.

In MTRL, estimating sharing of behaviors from policy $\pi_j$ to $\pi_i$ depends on the task at hand Task $i$, the environment state $s$, and the behavior proposal of the other policy at that state $\pi_j(s)$. Therefore, we must identify shareable behaviors in a task and state-dependent way, being aware of how all the task policies $\pi_j$ change over the course of training. For example, two task policies, such as Boil-Water and Make-Salad in Figure 1, may share only a small segment of behavior or may initially benefit from shared exploration of a common unknown environment. But eventually, their behaviors become conflicting or irrelevant to each other as the policies diverge into their own task-specific behaviors.

**Q-switch**: We propose to utilize each task's learned Q-function to evaluate shareable behaviors. The Q-function, $Q_i(s, a)$, of Task $i$ estimates the expected discounted return of the policy after taking action $a$ at state $s$ (Watkins & Dayan, 1992). Although this is an estimate acquired during training, it is a critical component in many state-of-the-art RL algorithms (Haarnoja et al., 2018; Lillicrap et al., 2015). It has also been used as a filter for high-quality training data (Yu et al., 2021; Nair et al., 2018; Sasaki & Yamashina, 2020), suggesting the Q-function is effective for evaluating and comparing actions during training. Thus, we use the Q-function as a switch that rates action proposals from other tasks' policies for the current task's state $s$. While the Q-function could be biased when queried with out-of-distribution actions from other policies, we will explain how this is corrected in practice in the next section on how the Q-switch is used in behavior sharing. Thus, this simple and intuitive function is state and task-dependent, gives the current best estimate of which behaviors are most helpful (those with high Q-values) and conflicting or irrelevant behaviors (those with low Q-values), and is quickly adaptive to changes in its own and other policies during online learning.

### 4.3 Incorporating Shareable Behaviors

We propose to use other task policies as behavioral suggestions to aid the exploration of the current task. This enables us to incorporate helpful behaviors from other tasks without assuming access to a reward re-labeler. Furthermore, this allows the agent to observe the effect of a proposed behavior in the current task, so only the behaviors with high task rewards are effectively incorporated into the task policy through the training process. This makes our sharing mechanism applicable to general task families, including those with eventually conflicting optimal behaviors.

**Mixture of Policies**: To allow for selective behavior sharing, we use a mixture of all task policies to gather training data for each task. Training a mixture of policies is a popular approach in hierarchical RL (Çelik et al., 2021; Daniel et al., 2016; End et al., 2017; Goyal et al., 2019) to attain reusable skills. In MTRL, we aim to benefit similarly from reusable behaviors. The main differences are that each policy is specialized to a particular task and the mixture is only used to gather exploratory data.

To this end, we define a mixture policy $\pi_i^{mix}(a|s)$ for each task $i$ over all the task policies $\pi_j$. At each timestep, $\pi_i^{mix}$ uses Q-switch to choose the best-scored policy $\pi_j^*$ at any state and samples an action from that policy, $a \sim \pi_j^*$ (Figure 2). This mixture allows us to activate multiple policies at different states in an episode, so we can selectively incorporate shareable behaviors from various tasks in a task and state dependent way. And while there are more sophisticated ways of scoring and incorporating policy proposals, such as defining a probabilistic mixture, we found that our simple method QMP works well in practice while requiring minimal hyperparameter tuning or computational overhead.

Importantly, the mixture of policies is used only to gather training data and is not trained directly as a common policy. Instead, each task policy $\pi_i$ is trained with data gathered for its own task $T_i$ by the mixture of policies $\pi_i^{mix}$ with the assistance of $Q_i$ as a switch (see Figure 2). The only difference to conventional RL is that the training dataset is generated by $\pi_i^{mix}$ and not $\pi_i$ directly, thus benefiting from shared behaviors through exploration without being limited by tasks with conflicting behaviors.

Together, the Mixture of Policies, $\pi_i^{mix}$ enables multi-task exploration guided by Q-switch, the current task's objective. While it is possible for the Q-switch to choose a harmful action due to error on an out-of-distribution action, it still helps exploration analogous to Q-learning. The agent observes the environment rewards after taking this action and will update its Q-function. Thus, any helpful behaviors seen in exploration are incorporated into $\pi_i$ through policy optimization. In contrast, conflicting or irrelevant behaviors, which represent an error in $Q_i$'s estimation, are used to update and correct $Q_i$ and thus the Q-switch. In addition, Q-switch starts as a uniform mixture but develops a stronger preference for policy $\pi_i$ as it becomes more proficient in task $i$. Consequently, cross-task behavior-sharing naturally decreases as $\pi_i$ specializes and requires less exploration.

## 5 Experiments

### 5.1 Environments

To evaluate our proposed method, we experiment with multi-task designs in navigation and manipulation environments shown in Figure 3. When evaluating the effectiveness of selective behavior sharing, the complexity of these multi-task environments is determined not only by individual task difficulty, more importantly, by the degree of similarity in behaviors between tasks. Thus to create challenging benchmarks, we ensure each task set includes tasks with either conflicting or irrelevant behavior. Further details on task setup and implementation are in Appendix Section A.2.

**Multistage Reacher:** The agent is tasked to solve 5 tasks of controlling a 6 DoF Jaco arm to reach multiple goals in an environment simulated in the MuJoCo physics engine (Todorov et al., 2012). In 4 out of the 5 tasks, the agent must reach 3 different sub-goals in order with some coinciding segments between tasks. In the 5th task, the agent's goal is to stay at its initial position for the entire episode. The observation space does not include the goal location, which must be figured out from the reward. Thus, for the same states, the 5th task directly conflicts with all the other tasks.

**Maze Navigation:** In this environment, the point mass agent has to control its 2D velocity to navigate through the maze and reach the goal, where both start and goal locations are fixed in each task. The observation consists of the agent's current position and velocity. But, it lacks the goal location, which should be inferred from the dense reward based on the distance to the goal. Based on the environment proposed in Fu et al. (2020), we define 10 tasks with different start and goal locations. The optimal paths for different tasks have segments that coincide and that directly conflict.

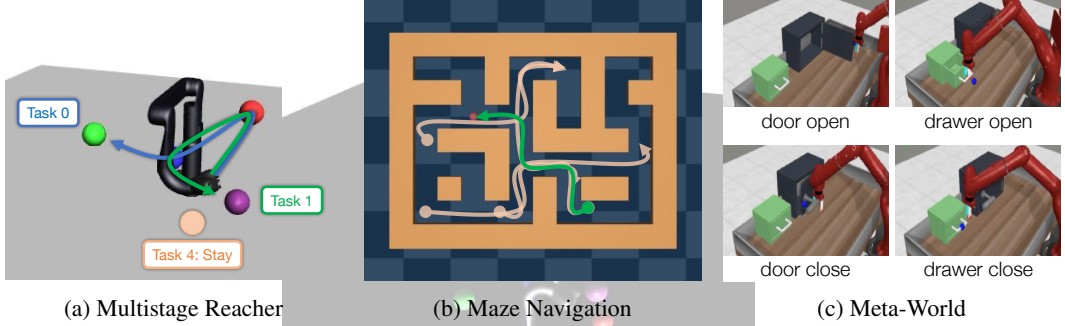

(a) Multistage Reacher      (b) Maze Navigation      (c) Meta-World

Figure 3: **Environments & Tasks**: (a) Multistage Reacher. The agent must reach 3 ordered subgoals in each task except Task 4, where the agent must stay at its initial position. See Appendix Table 1 for goal locations. (b) Maze Navigation. The agent (green circle) must navigate through the maze to reach the goal (red circle). Example paths for 4 other tasks are shown in orange. (c) Meta-World Manipulation. Consisting of tasks: door open, door close, drawer open, drawer close.

**Meta-World Manipulation:** We follow the modified 4-task shared-space setup of the Meta-World environment (Yu et al., 2019) proposed in Yu et al. (2021). It places the door and drawer objects next to each other on the same tabletop so that all 4 tasks (door open, door close, drawer open, drawer close) are solvable in a simultaneous multi-task setup, making it amenable to our problem domain (unlike the original MT10 task set which is built over separate environments). The observation space consists of the robot's proprioceptive state, the drawer handle state, the door handle state, and the goal location. While there are no directly conflicting behaviors between tasks, there are irrelevant behaviors. Thus, policies should learn to share behaviors when interacting with the same object while ignoring irrelevant behavior from policies that only interact with the other object.

## 5.2 Baselines

We used Soft Actor-Critic (SAC) Haarnoja et al. (2018) for all models. We first compare different forms of cross-task behavior-sharing in isolation from other forms of information-sharing. Then, we show how behavior-sharing complements parameter-sharing. For the non-parameter sharing version, we use the same architectures and SAC hyperparameters for policies across all baselines.

- **No-Shared-Behaviors** consists of $T$ RL agents where each agent is assigned one task and trained to solve it without any behavior sharing with other agents. In every training iteration, each agent collects the data for its own task and uses it for training.

- **Fully-Shared-Behaviors** is a single SAC agent that learns one shared policy for all tasks, which outputs the same action for a given state regardless of task (thus naturally does parameter sharing too). For the fairness of comparison, we adjusted the size of the networks, batch size, and number of gradient updates to match those of other models with multiple agents.

- **Divide-and-Conquer RL (DnC)** (Ghosh et al. (2018)) uses an ensemble of $T$ policies that shares behaviors through policy distillation and regularization. We modified the method for multi-task learning by assigning each of the policies to a task and evaluating only the task-specific policy.

- **DnC (Regularization Only)** is a no policy distillation variant of DnC we propose as a baseline.

- **UDS (Data Sharing)** proposed in Yu et al. (2022) shares data between tasks, relabelling with minimum task reward. We modified this offline RL algorithm for our online set-up.

- **QMP (Ours)** learns $T$ policies sharing behaviors via Q-switch and mixture of policies.

For further details on baselines and implementation, please refer to Appendix Section A.4.

## 6 Results

We conduct experiments to answer the following questions: (1) How does our method of selectively sharing exploratory behaviors compare with other forms of behavior sharing? (2) How crucial is adaptive behavior sharing? (3) Can QMP effectively identify shareable behaviors? (4) Is behavior sharing complementary to parameter sharing?

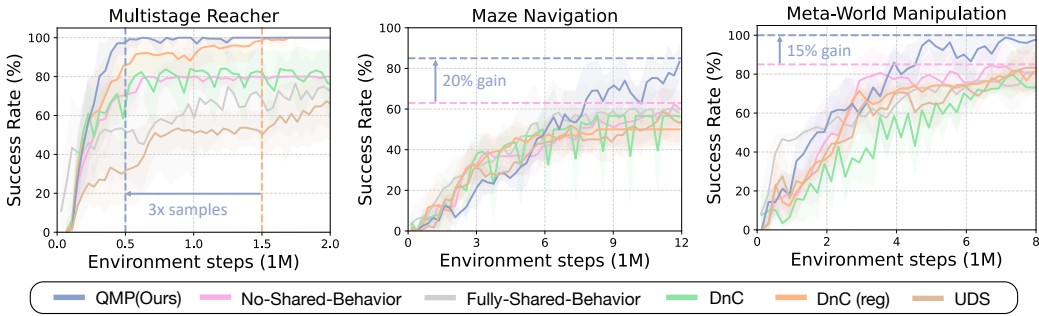

Figure 4: Comparison of average multitask success rate, over 10 evaluation episodes per task and 5 seeds for each method. The dashed lines highlight the gains of our proposed method (QMP) over the best baseline. QMP outperforms the baselines in terms of the rate of convergence (3x in Multistage Reacher) and the task performance (20% in Maze Navigation and 15% Meta-World Manipulation).

## 6.1 Baselines: How exploration sharing compares to other forms of behavior sharing?

To verify QMP's efficacy as a behavior sharing mechanism, we compare against several behavior sharing baselines on 3 environments: Multistage Reacher (5 tasks), Maze Navigation (10 tasks), and Meta-World Manipulation (4 Tasks) in Figure 4. Overall, QMP outperforms other methods in terms of sample efficiency and final performance across all task sets.

In Multistage Reacher, our method reaches 100% success rate at 0.5 million environment steps, while DnC (reg.), the next best method, takes 3 times the number of steps to fully converge. The rest of the methods fail to attain the maximum success rate. The UDS baseline performs the worst, illustrating that data sharing can be ineffective without ground truth rewards. The Fully-Shared-Behaviors baseline performs similarly, highlighting the challenge of the conflicting behaviors between tasks.

In the Maze Navigation environment, we test the scalability of our method to a larger 10-task set. QMP successfully solves 8 tasks out of 10, while other methods plateau at around a 60% success rate.

In the Meta-World Manipulation environment, our method reaches almost 100% success rate after 8 million environment steps while other methods plateau at around 85%. This is significant because this task set contains a majority of irrelevant behavior: between policies interacting with the door versus the drawer and between pulling on the object handles versus pushing. The fact that QMP still outperforms other methods validates our hypothesis that shared behaviors can be helpful for exploration even if the optimal behaviors are different. We note that the Fully-Shared-Behaviors baseline performs very well initially but quickly plateaus. The initial performance is likely due to the shared policy also benefiting from shared parameters across tasks, whereas the other methods learn separate networks for each task. However, the tasks eventually diverge based on the object to be manipulated, making full behavior-sharing suboptimal.

## 6.2 Ablations: How crucial is adaptive behavior sharing?

We look at the importance of an adaptive, state-dependent Q-switch by comparing QMP to two ablations where we replace the Q-switch with a fixed sampling distribution over task policies to select which policy is used for exploration.

- **QMP-Uniform** replaces the Q-switch with a uniform distribution over policies to verify the importance of selective and adaptive behavior sharing.

- **QMP-Domain-Knowledge** replaces the Q-switch with a hand-crafted, fixed policy distribution based on domain knowledge of the relationship between tasks (i.e., sampling probabilities proportional to the number of shared sub-goal sequences between tasks in Multistage Reacher, see Appendix A.2 for details) to verify the importance of adaptive and training-progress aware behavior sharing.

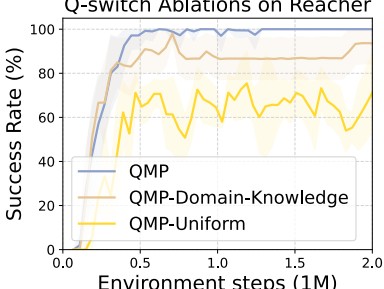

Figure 5: An adaptive state and task dependent Q-switch is crucial.

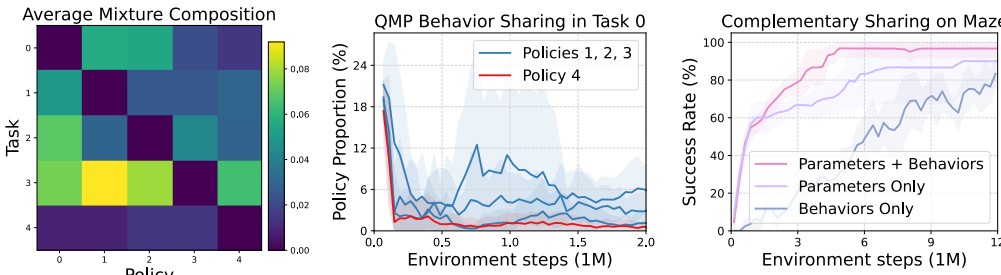

(a) Cross-task sharing proportion (b) Behavior-sharing over training (c) Behavior & parameter sharing

Figure 6: (a) Proportion of shared behavior on Reacher Multistage averaged over training: Each cell (row $i$, col $j$) represents sharing contribution of Policy $j$ for Task $i$ (diagonal zeroed out for contrast). (b) Mixture probabilities of other policies over the course of training for Task 0 in Multistage Reacher: behavior-sharing decreases as $\pi_0$ improves. Q-switch shares the least from the conflicting task Policy 4, shown in red. Full analysis for all tasks in Appendix Figure 9. (c) Combining our proposed method with parameter sharing using a multi-head shared architecture (in pink) outperforms both components on their own: SAC with shared parameters (purple) and our method without shared parameters (blue).

In Figure 5, we see QMP-Uniform reaches around 60% success rate, lower than the worst performing behavior sharing baseline, demonstrating that a poor choice of Q-switch can significantly hinder learning in our framework. Uniformly randomly using other task policies for exploration can inject a significant amount of low-reward data, making learning inefficient.

For QMP-Domain-Knowledge, we assign the probability of selecting $\pi_j$ for Task $i$ by the number of shared sub-goal sequences between Tasks $i$ and $j$. QMP-Domain performs well initially but plateaus early. An improvement over QMP-Uniform shows the importance of task-dependent behavior sharing, while the performance deficit from QMP suggests that state-dependent and training-adaptive sharing is necessary. Crucially, defining such a specific domain-knowledge-based mixture of policies is generally impractical and requires knowing the tasks beforehand. While we specifically designed the Multistage Reacher for this didactic analysis, such domain knowledge is exponentially harder to define for complex tasks, especially if we want a state-dependent mixture.

### 6.3    Can QMP effectively identify shareable behaviors?

Figure 6a analyzes the effectiveness of the Q-switch in identifying shareable behaviors by visualizing the average proportion that each task policy is selected for another task over the course of training in the Multistage Reacher environment. This average mixture composition statistic intuitively analyzes whether QMP identifies shareable behaviors between similar tasks and avoids behavior sharing between conflicting or irrelevant tasks. As we expect, the Q-switch for Task 4 utilizes the least behavior from other policies (bottom row), and Policy 4 shares the least with other tasks (rightmost column). Since the agent at Task 4 is rewarded to stay at its initial position, this behavior conflicts with all the other goal-reaching tasks. Of the remaining tasks, Task 0 and 1 share the most similar goal sequence, so it is intuitive why they benefit from shared exploration and are often selected by their respective Q-switches. Finally, unlike the other tasks, Task 3 receives only a sparse reward and therefore relies heavily on shared exploration. In fact, QMP demonstrates the greatest advantage in this task (Appendix Figure 8). Furthermore, we see that total behavior sharing decreases throughout training in all tasks (Figure 6b), which demonstrates a naturally arising preference in the Q-switch for its own task-specific policy as it becomes more proficient.

We qualitatively analyze behavior sharing by visualizing a rollout of QMP during training for the Drawer Open task in Meta-World Manipulation (Appendix Figure 10). We see that it switches between all task policies as it approaches the drawer, uses drawer-specific policies as it grasps the handle, and opening-specific policies as it pulls the drawer open. In conjunction with the overall results, this supports our claim that QMP can effectively identify shareable behaviors between tasks.

### 6.4    Is behavior sharing complementary to parameter sharing?

It is important that our method is compatible with other forms of multitask reinforcement learning that share different kinds of information, especially parameter sharing, which is very effective under

low sample regimes (Borsa et al., 2016; Sodhani et al., 2021) as we saw in the initial performance of Fully-Shared-Behaviors in Meta-World Manipulation. While we use completely separate policy architectures for previous experiments to isolate the effect of behavior sharing, QMP is flexible to any design where we can parameterize $T$ task-specific policies. A commonly used technique to share parameters in multi-task learning is to parameterize a single multi-task policy with a multi-head network architecture. Each head of the network outputs the action distribution for its respective task. We can easily run QMP with such a parameter-sharing multi-head network architecture by running SAC on the multi-head network and replacing the data collection policy with $\pi_i^{mix}$.

We compare the following methods on the Maze Navigation environment in Figure 6c.

- **Parameters Only**: a multi-head SAC policy sharing parameters but not behaviors over tasks.

- **Behaviors Only**: Separate task policy networks with QMP behavior sharing.

- **Parameters + Behaviors**: a multi-head SAC network sharing behaviors via QMP exploration.

We find that sharing Parameters + Behaviors greatly improves the performance over both the Shared-Parameters-Only baseline *and* Shared-Behaviors-Only variant of QMP. This demonstrates the additive effect of these two forms of information sharing in MTRL. The agent initially benefits from the sample efficiency gains of the multi-head parameter-sharing architecture, while behavior sharing with QMP accelerates the exploration via the selective mixture of policies to keep learning even after the parameter-sharing effect plateaus. This result demonstrates the compatibility between QMP and parameter sharing as key ingredients to sample efficient MTRL.

# 7  Limitations

When we have prior knowledge that a task set has little or no conflicting behaviors, methods like DnC, with uniform behavior sharing and a tunable hyperparameter governing the strength of sharing, can fully leverage shared behavior between tasks and work very efficiently. In contrast, QMP does not assume a-priori that behaviors are fully shareable and therefore shares behavior selectively and adaptively as it learns the tasks. As a result, it can be more conservative in the amount of shared behavior and, thus, less sample efficient compared to methods specific to these task families. Essentially, QMP is a robustly efficient MTRL method applicable to a variety of task sets, but trades off on absolute performance on either end of the spectrum (i.e., fully non-conflicting or

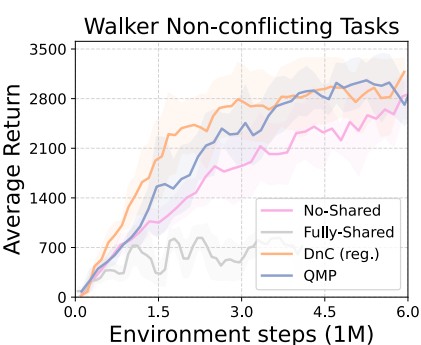

Figure 7: QMP shares conservatively.

fully conflicting tasks). We found this to be the case in a multi-task set where a Walker agent learns different gaits with no directly conflicting behaviors like walking, balancing, and crawling: QMP still outperforms no shared behavior or fully shared behavior baselines but DnC (Reg. only) works best (see Figure 7 and Appendix Section A.3.3). However, in task sets with potentially conflicting behaviors or where the similarity in task behaviors is not known, we believe QMP to be the best option for robust multi-task behavior sharing as demonstrated in Section 6.1.

# 8  Conclusion

We introduce the problem of selective behavior sharing to improve exploration in MTRL for tasks requiring differing optimal behaviors. We propose Q-switch Mixture of Policies (QMP), that incorporates behaviors between tasks for exploration through a value-guided selection over behavior proposals. Experimental results on manipulation and navigation tasks demonstrate that our proposed method effectively learns to share behavior to improve the rate of convergence and task performance in task families even with conflicting or irrelevant behaviors, which highlights the importance of *selective* behavior sharing. We further show that our method is complementary to parameter sharing, a popular MTRL strategy, demonstrating the effectiveness of behavior sharing in conjunction with other forms of information sharing in MTRL. Promising future directions include extending exploration improvements of selective behavior sharing problems where tasks are not necessarily learned simultaneously, such as transfer learning and continual learning in RL.

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
