# OpenReview forum: "Efficient Multi-Task Reinforcement Learning via Selective Behavior Sharing"
_NeurIPS.cc/2023/Conference — Submitted to NeurIPS 2023_

### Official Review · Reviewer_uAJb · 2023-06-21

**Soundness:** 3 good
**Presentation:** 3 good
**Contribution:** 2 fair
**Rating:** 5
**Confidence:** 3

**Summary:**

This work presents Q-switch Mixture of Policies (QMP) that identifies shareable behaviors and incorporates shareable behaviors. The authors propose to utilize each task’s learned Q-function to evaluate shareable behaviors, and incorporate helpful behaviors from other tasks to aid the exploration of the current task. Experiments on manipulation and navigation tasks are done to validate the proposed method.

**Strengths:**

1, The paper is written well and the MTRL framework has the potential to generalize to kinds of RL tasks because of its simplicity. The idea of Q-switch is straightforward, but seems to work in the experiments.

2, The analysis is comprehensive, and validates the effectiveness of the behaviors identifying and incorporating. Moreover, behavior sharing seems to be a good complement to parameter sharing. By combining them, the sample-efficiency would get improved further.


**Weaknesses:**

Though compared with many benchmarks, the experiment environments are sort of over-simplified. I would suggest testing the framework in the meta-world environment with more tasks, like insert peg, pick&place, to make the results more convincing and reliable.

**Questions:**

In figure2, the value of Q(s,a_3) is highest, but why is policy 1 chosen in data gathering? Besides, in section 3, action space S -> action space A.

**Limitations:**

As discussed above in the weaknesses, it's still a good paper to accept.

---

> ### Author Rebuttal · Authors · 2023-08-10
>
> Thank you for your thoughtful review and feedback that helped us improve the paper. We address each of your concerns below, including new experiments.
>
> ### More Complex Environment added: Metaworld MT10
> As advised, we have added an MT10 experiment to test QMP’s contribution to parameter-shared and parameter-separated policy architectures in Rebuttal Fig. P6.
>
> Discussion: **Metaworld state space is not ideal for sharing.**
> - As noted in Sec 5.1, we choose the 4-task setup of Yu et al. (2021) [3] to ensure consistent state space. It is important to recognize that MT10’s task set is peculiarly constructed over separate environments that are *artificially connected* through overloaded state dimensions for different objects. This makes the problem of behavior-sharing inapplicable because there is no shared state structure to generalize the behaviors over.
> - In fact, we find that **even parameter-sharing is not a good solution to MT10**, despite Meta-world paper reporting its best results on parameter-sharing SAC. Our results in Rebuttal Fig. P6 that while parameter-sharing accelerates learning at the start, it converges to a suboptimal value of 68.3% success rate (our numbers match Metaworld paper). In contrast, naively training parameter-separate SAC policies reach around 75% success rate! Crucially, when we add QMP’s behavior sharing, it improves the success rates to over 80% while also improving sample efficiency.
> - Since parameter-sharing makes learning unstable, likely due to gradient interference between tasks. Thus, when QMP is combined with parameter-sharing, it also suffers from similar gradient interference.
>
> ### The experiments are computationally complex because of online multi-task RL from scratch
> Since we do not assume demonstration data, most of our experiments take 1.5 days to train on an RTX 3090, as all the tasks need to be trained with SAC. Some MT10 experiments we now added took about **7 days to complete** per run! Given the state of RL methods and our non-industry scale compute budget, we find this to be the most feasible environment setup we can train on, while still being able to perform exhaustive experiments on 5 seeds over 6 baseline methods. Such computational complexity is precisely why behavior-sharing methods such as QMP are important, to improve the sample efficiency in online multi-task RL.
>
> ### Clarifying Fig. 3
> As you rightly interpret, the Gather data step is indeed done by Policy 3. We intended to show this by the robot pose matching policy 3’s action proposal. To eliminate any confusion, we have now labeled the best behavior with “$a_3$” to show that policy 3 was chosen for data gathering. We also hope the added Algorithm 1 makes it easier to follow our approach.
>
> Please let us know if there are any other concerns we can address to help increase your score.

---

> > ### Comment · Reviewer_uAJb · 2023-08-17
> >
> > Thanks for your responses. I totally understand the adaptation issues and computational power concerns. I may keep my score for now.

---

### Official Review · Reviewer_cxcr · 2023-06-22

**Soundness:** 2 fair
**Presentation:** 3 good
**Contribution:** 2 fair
**Rating:** 4
**Confidence:** 4

**Summary:**

The paper studies sharing behaviors between tasks in multi-task reinforcement learning. In the proposed method, each task maintains an independent policy network. During online exploration, it selects the action maximizing the task's Q function among actions proposed by all the tasks' policies. Experimental results show the method improves multi-task performance in three continuous control environments.

**Strengths:**

1. The paper is well written. The motivation of behavior sharing for MTRL is clear.
2. In experiments, figures and results are clearly presented.


**Weaknesses:**

1. The paper makes a strong assumption that tasks are only different in reward functions. Many complementary methods, like parameter sharing, tackle a wide problem setting where the transition functions and state spaces can be diverse.
2. In baseline methods: the Fully-Shared-Behaviors baseline, a policy without any task information as input for multi-task RL, is weird. A fully-shared baseline with task identifier input makes more sense.
Two ablation methods seems unnecessary, since the proposed method is simple enough.
3. In experimental results, the proposed method does not outperform baselines very significantly.

**Questions:**

As shown in figure 6.c, parameter sharing method contributes more than the proposed method. The proposed method+parameter sharing only improves a little to the method sharing parameters only. How are the results of parameters+behaviors sharing in the other two environments?

**Limitations:**

Authors discussed some of the limitations and they should be addressed in future work.

---

> ### Author Rebuttal · Authors · 2023-08-10
>
> Thank you for your thoughtful and constructive feedback and suggestions for additional parameter sharing experiments which we have added.  We address each of your concerns below.
>
> ### Shared state space and transition function
> As we address in more detail in the general response, shared state space and transition function are common assumptions to many prior works [1-7] and we believe to be an important subset of MTRL problem setups. Furthermore, as you mention, parameter sharing is a complementary approach and can be combined with and does not detract from our work in MTRL settings with a shared environment. We empirically elaborate below that parameter-sharing’s applicability to a method does not mean it necessarily improves learning and can even hurt. In that sense, behavior-sharing can also be “applied” to different environments. But it is more likely to help when tasks are conducted in the same environment, which is true for a large family of single-agent RL settings.
>
> ### Fully-Shared-Behaviors Baseline
> This baseline in Section 6.1 answers the question “How exploration sharing compares to other forms of behavior sharing?”. The Fully-Shared-Behaviors baseline represents one end of the spectrum of behavior sharing which would be ideal when the agent’s optimal behaviors across tasks do not conflict at all. Providing a task identifier as input makes this a parameter-sharing approach, because except the task ID the parameters of all task policies are shared. We do already evaluate a multi-head shared SAC policy (equivalent to a shared policy with task ID) in Figure 6c and show that the benefits of parameter sharing are complementary with behavior sharing.
>
> ### Parameter Sharing vs. Behavior Sharing
> As suggested, we have added the QMP + parameter sharing experiments in Rebuttal Fig. P4-P6. We thank the reviewer for this suggestion and discuss the crucial insights about parameter-sharing below:
> - **Parameter-sharing is suboptimal with conflicting tasks**: In Maze (Paper Fig 6), Reacher (Fig. P4) and new MT10 results (Fig P6), parameters-sharing SAC converges suboptimally. Especially, in Reacher where Task 4 is behaviorally different (stay at the same position) from other tasks (reach subgoals), parameter-sharing is highly suboptimal (60%). On closer inspection, we find that because Task 4 is easier to learn, parameter-sharing often converges suboptimally to that task which hurts its learning on other tasks. Crucially, parameter-sharing is even worse than no-sharing at all on Reacher (60 v/s 80%) and MT10 (68% v/s 75%), which shows that **parameter-sharing can hurt performance**. Similar phenomenon has been observed in prior works [R3-1].
> - **QMP can help deal with conflicting tasks**: QMP + parameter-sharing on Maze and Reacher improves the performance over the parameters-only results. In Reacher, the influence of parameter-sharing is so negative that QMP-separate is better than QMP-parameter-sharing. But QMP consistently improves the performance over no-parameter-sharing in all 4 environments we tested: Reacher, Maze, MT4, and MT10. In MT10, QMP + no-parameter-sharing achieves 85% success as compared to Metaworld paper’s reported results of 68.3% with parameter-sharing. While there exist other approaches that can help improve MT10 performance, our claim is simply that when the influence of parameter-sharing is disentangled, behavior-sharing is consistently shown to help.
> - **QMP + parameter-sharing is not always complementary**. While we could not hyperparameter-tune sufficiently in MT4 and MT10 experiments given the short timeframe of rebuttal, our preliminary results show that the combination of QMP with parameter-sharing can negatively interfere. Parameter-sharing makes QMP worse on Reacher (Fig P4) while QMP makes parameter-sharing worse on MT4 and MT10 (Fig P5-P6). We hope stabilizing parameter-sharing with approaches such as gradient surgery [R3-1] can improve the performance of QMP’s combination with it, just like QMP consistently helps over no-parameter-sharing. However, given the open challenges of parameter-sharing itself (as demonstrated above), this is beyond the scope of our work’s focus on behavior sharing.
>
> ### Significance of Experimental Results
> QMP employs a simple hyperparameter-free approach of utilizing a Q-function to share behaviors without introducing any additional method-specific hyperparameter tuning. We demonstrate 3x sample efficiency improvement in Reacher and 15-20% optimal performance gain in Maze and Metaworld tasks over all existing approaches. In Fig 4 (right) and Fig. 6 (c), QMP is the only method that achieves a 100% success rate, while the best baselines show a suboptimal performance of 85% and 90%. Finally, in newly added MT10 results in Fig P6, QMP helps achieve success rates of 85% while parameter-sharing converges at 68%. We hope the simplicity of QMP without the need for new hyperparameters, and consistent performance improvements are valuable.
>
> ### Need for ablations
> The ablations demonstrate the need for an adaptive behavior sharing scheme via our Q-filter, by comparing against fixed schemes of behavior-sharing, including a manually crafted and uniform sharing scheme. As Reviewer R1 pointed out, it is important to assess the question of static task similarity and whether that itself is enough to perform behavior-sharing. We would be happy to add any additional ablations that the reviewer thinks are necessary to add in place of or in addition to these ablations?
>
> ### [References]
> [R3-1] Yu, Tianhe, et al. "Gradient surgery for multi-task learning." NeurIPS 2020.
>
> We hope this clarifies and addresses the concerns raised. We again thank the reviewer for the new insights derived on parameter-sharing and we would be happy to answer further questions, if any remain.

---

### Official Review · Reviewer_xBdd · 2023-06-30

**Soundness:** 2 fair
**Presentation:** 3 good
**Contribution:** 2 fair
**Rating:** 5
**Confidence:** 4

**Summary:**

The paper introduced a new exploration mechanism for MTRL. They suggested training a different policy for each task and “sharing the behaviors” between them. In order to do so, each policy, in a certain state, chooses the most suitable action using its own Q function. The author evaluates several MTRL benchmarks and shows increased sample efficiency and final performances over behavior-sharing baselines.

**Strengths:**

$\underline{\text{Clarity:}}$

1. Overall, the paper is coherent and easy to follow
2. The introduction is well-written and the motivation for the work is clear

$\underline{\text{Significance:}}$

1. The method seems quite general and might be useful in many cases

2. To the best of my knowledge the idea of using the Q function as a metric for policy selection is novel

**Weaknesses:**

My main issue is with the technical soundness of the paper. Throughout the paper, the level of the formulation was relatively low, and I spotted some inaccuracies in notation and claims. In general, I understood the motivation for the Q-switch, but there lacks some theoretical analysis or empirical study to support it. I think this method might be suited for some set of tasks, but probably have a limitation, due to the generalization capabilities of the Q function, that was not discussed in the paper. Here are some more specific examples regarding the technical quality of the paper:

$\underline{\text{In the problem formulation section:}}$

1. The MDP components are not defined. Are the state and action spaces continuous or discrete? Should state that $\gamma \in \mathbb{R}$

2. In line 104 you state that $T$ is a number of tasks in the task set and in line 109 you denote the i’th task in the set as $T_i$. This is a confusing abuse of notation.

3. In line 107 - “We parameterize the multi-task solution as…” what does the solution for a multi-task mean?

4. In line 109+110 - the objective is not formulated. “the tasks are uniformly sampled during training” - what does that mean? From which distribution the tasks are being sampled? Does the sample accure at the beginning of the training phase once?

5. In line 112 - what is $\pi_i^*$? It is not defined.

$\underline{\text{In section 4.3:}}$

1. line 173 - “over all the task policies $\pi_j$” -> “over all the task policies $\left[\pi_j\right]_{j=1}^T$”

2. In line 173 - how does $\pi_i^*$ defined? I believe it is an abuse of notation from section 3.

$\underline{\text{Related work and baselines:}}$

1. Limited coverage of skill learning and intrinsic reward literature. I don’t think the statement in line 101 (“.. assume that the optimal behaviors of different tasks do not conflict with each other”) is true for many skill-learning methods

2. Although this work approach is quite different than intrinsic reward/skill learning, I believe that a standard state visitation bonus should be competitive (or at least a good baseline) for the evaluation benchmarks

$\underline{\text{Experiments:}}$
1. In section 6.2, the chosen baselines (QMP-uniform and QMP-domain knowledge) are too simplistic, please provide a more solid baseline, e.g. the one you suggested in line 178 (a probabilistic mixture).

2. In Figure 6(c) you show that the best performances were achieved when incorporating both parameter sharing and behavior sharing, and showed that using only parameter sharing beats using only behavior sharing. This raises the question of what would happen if we used different kinds of exploration mechanisms together with your method (e.g. intrinsic exploration). Overall, I feel that this evaluation is limited, both in the variation in testing environments and exploration combinations.

**Questions:**

1. Typo - line 106: action space $\mathcal{S}$ -> action space $\mathcal{A}$
2. Can you clarify if the set of policies that $\pi^{mix}_i$ is defined over includes $\pi_i$?
3. Line 76 - “We do not require pre-defined task similarity or exploration bonus” - can you clarify this?
4. In which task families do you believe your approach will have the upper hand over simple exploration bonuses (e.g. state visitations)?
5. Can you provide a pseudo-code for your algorithm?
6. Can you provide a version of the graphs in Figure 4 with more env-steps, I find it quite surprising that the no-shared behavior baseline doesn’t converge to the optimal performances.
7. Can you provide a graph for the phenomenon discussed in lines 192-193? Something like the rate the Q-switch of task $i$ chooses $\pi_i$ as a function of the number of train env steps.
8. In Figure 6(a), doesn’t every row should sum to 1? I’m not entirely sure I understood this figure.

**Limitations:**

Although the authors raised a valid point in the Limitation section of the paper, I believe other limitations of the method exist and aren’t addressed (please see the weaknesses section for further details).

---

> ### Author Rebuttal · Authors · 2023-08-10
>
> Thank you for your rigorous, thoughtful and constructive feedback on the technical soundness of the Q-switch and suggestions for additional baselines. We address the main concerns below and the questions after.
>
> ### Technical Soundness of Q-switch
> While theoretical guarantees are outside of the scope of this paper, we believe there is ample empirical evidence to support a Q-function based Q-switch. As we note in Section 4.2, Q-functions have been widely used for imitation learning and offline RL (Yu et al., 2021; Nair et al., 2018; Sasaki & Yamashina, 2020) to filter for high-quality data, which provides evidence for its effectiveness for evaluating actions from other policies. We performed empirical studies on the Q-switch in our paper showing that it:
> - successfully identifies shareable behaviors (Fig 6a)
> - becomes more selective over the course of training (Fig 6b)
> - learns to not share from conflicting tasks (Supp. Fig 9)
> - effectively switches over different policies within an episode (Supp. Fig. 10)
> Due to the prior work and our own empirical findings, we believe that our proposed method is well-supported.
>
> The reviewer also correctly points out that Q-switch can be limited by error in the Q-function, which we already discussed in Section 4.3 (line 185). We reiterate that Q-switch has a **self-correcting** characteristic: if the Q-function generalizes poorly and the Q-switch chooses some non-optimal action, the agent will get data, and train its Q-switch — similar to how standard Q-learning corrects its Q-function by exploring and making mistakes.
>
> ### Potentially Related Work
> - **Skill-learning**: We appreciate the reviewer’s suggestion, however, skill learning can broadly refer to anything from single-task RL to hierarchical RL. To the best of our knowledge, we do not miss any relevant baselines for MTRL behavior-sharing. We would greatly appreciate some references of papers that the reviewer identifies as missing and are happy to discuss them in the revision.
> - **Exploration / Intrinsic Reward**
>   + We highlight that our evaluation benchmarks are already dense-reward tasks (except one task in Reacher), thus, specialized exploration is not expected to help in these tasks. In response to the reviewer’s comment, “I believe that a standard state visitation bonus should be competitive”, we add a new experiment in Rebuttal Fig. 2, where SAC with increased exploration bonus (by increasing the entropy coefficient) does not benefit in performance.
>   + Intrinsic rewards help exploration in tasks with sparse rewards and can be complementary to QMP. The challenge we address in this work is not that individual tasks are hard to learn, but how to learn multiple related tasks together efficiently. The key idea of behavior-sharing is that it *shortcuts the need to explore* by exploiting similar experiences made in other tasks.
>   + An investigation of combinations with different explorations mechanisms is tangential to the focus of this work, but would be an interesting extension. For example, a sparse reward multi-task problem would likely require both multi-task behavior sharing and intrinsic exploration for each task.
>
> ### Potential Baselines
> - **More solid baselines** Section 5.2 and 6.1 have solid baselines including all prior approaches in behavior sharing. Section 6.2 particularly ablates the need for an adaptive behavior sharing scheme via our Q-filter, by comparing against a manually crafted and uniform sharing scheme under our method. We would appreciate it if the reviewer can list down any other baselines or ablations from prior work they believe are missing.
> - **Probabilistic mixture** A probabilistic mixture of policies is a design choice of our approach where arg-max is replaced with softmax. However, in our initial experiments, we found no significant improvement in performance and it came with an additional hyperparameter of tuning the temperature coefficient. We attach that result in the Rebuttal Fig. P3 to justify the design choice of argmax over softmax due to its reliable performance and simplicity.
>
> ### Questions
> We greatly appreciate the detailed feedback on the typos and improper notations. We are glad it didn’t hinder the general understanding and flow of the method. We have fixed it in our manuscript thanks to the reviewer’s detailed comments.
> 1. Fixed typo.
> 2. $\pi_i^{mix}$ is a mixture of policies over all task policies including $\pi_i$; we will clarify in the paper.
> 3. Our method does not require any prior knowledge about the relationships between tasks like Kalashnikov et al. (2021b) does by assuming given “task-skill groupings”. We also do not use exploration bonuses like Bangaru et al. (2016).
> 4. While simple exploration bonuses are typically applied to a single task, our method improves exploration by sharing information between tasks. Therefore, our method would be more effective in task families with more shared behavior between tasks.
> 5. Thanks, we added pseudo code in Rebuttal Algorithm 1.
> 6. We have provided this graph in Rebuttal Fig. P7. The no-shared behavior baseline does converge to optimal performance but takes around 2x the number of samples to do so.
> 7. In Supp. Fig. 9 in the paper we already provide the proportion of shared behavior from each policy over training in the Multistage Reacher task and discuss how behavior sharing decreases over training as hypothesized in lines 192-193.
> 8.  Fig. 6a -- the caption notes that the diagonal is zeroed out. Yes, they should sum to 1 otherwise. So, the diagonal = 1 - (sum of non-diagonal row entries). We zero out the diagonal for color-scaling, because diagonal elements are on the order of 0.8 - 0.9 while non-diagonal elements are 0.0 - 0.1, so including the diagonal values removes the contrast we want to show between low-valued elements.
>
> We hope this clarifies and addresses the concerns raised and we would be happy to answer any further questions.

---

> > ### Comment · Reviewer_xBdd · 2023-08-19
> >
> > I appreciate the clarifications and additional results.
> > Your reply on the the Technical soundness of Q-switch relaxed my concern, especially the part regarding the self-correcting characteristic. I also appreciate the response regarding exploration and intrinsic rewards.
> > Regarding baselines, I mainly aimed for methods on the line of e.g Pertsch et al - Accelerating Reinforcement Learning
> > with Learned Skill Priors, CoRL 2020. There are many extensions to this line of work, some of which you referred to, but haven't discussed in details. Can you please explain why these are not valid baselines?
> >
> > Due to the clarifications and additional results, I have raised my score.

---

> > > ### Author Response · Authors · 2023-08-20
> > > **Adding discussion on skill-based RL (offline data; temporal abstraction) v/s MTRL (online RL; behavior-sharing)**
> > >
> > > We sincerely thank the reviewer for acknowledging our rebuttal and pointing out the skill-based RL works.
> > >
> > > The key premise of [Pertsch et al. 2020] and its extensions, often referred to as skill-based RL, is how to extract temporally extended behaviors from an **offline, task-agnostic dataset** of agent trajectories of *meaningful* interaction with environments (e.g., human-collected data). There are two phases: 1. Use the offline data to learn a skill space (where a skill is usually a sequence of N actions: $a_1, ... a_N$), and 2. On **one** downstream task, train a new policy whose action space is now the pretrained skill space. This allows efficient downstream learning in a sparse reward task as the effective horizon of the task is sharply reduced, thanks to learning meaningful skills from a good behavioral offline prior dataset.
> > >
> > > We actually consulted with the author of the prior work you listed, and the key conclusion was that these methods are not comparable, but rather complementary. Specifically, skill-based RL like [Pertsch et al. 2020] can learn a skill space and QMP can do **online multi-task RL** with this skill space as the new action space. So, QMP + Skills can perform **temporally extended behavior-sharing** between the multiple policies learned on top of a skill space! Creating new benchmarks on *long-horizon multi-task RL augmented with offline data* would be a very exciting future research direction! We thank the reviewer for this interesting perspective.
> > >
> > > To sum up, skill-based RL methods are not ideal baselines for our work because:
> > > - Our problem formulation of online MTRL does not assume access to any **dataset** with agent trajectories, unlike skill-based RL methods which tackle a totally different problem.
> > > - Skill-based RL methods mainly consider a **single target task**, while our proposed method aims to simultaneously learn **a set of multiple tasks** with improved overall efficiency.
> > > - Skill-based RL’s key focus is enabling a challenging sparse-reward task with temporal abstraction, while behavior-sharing methods, like ours, focus on selectively sharing action proposals from other tasks. As we show, our approach shows benefits in dense reward tasks too.
> > >
> > > Since offline datasets are a key requirement for such skill-based RL methods as [Pertsch et al. 2020], the closest baseline we can formulate is to compare with an approach that “consider the other tasks as unlabeled datasets for our current task”. This is **exactly** the UDS (Data Sharing) baseline [Yu et al. 2022] in our experiments, which we already show doesn’t work well for online MTRL.
> > >
> > > We will add the above insightful discussions to the revised paper: (a) the complementary nature of skill-based RL + MTRL behavior sharing and (b) trying to apply skill-based RL makes it equivalent to UDS. Thank you again for genuinely helping us improve our paper’s discussion, and we hope this would alleviate any leftover concerns.
> > >
> > > [Pertsch et al. 2020] Pertsch et al. "Accelerating Reinforcement Learning with Learned Skill Priors" CoRL 2020.
> > > [Yu et al. 2022] Yu et al. How to leverage unlabeled data in offline reinforcement learning. ICML 2022.

---

### Official Review · Reviewer_npBg · 2023-07-04

**Soundness:** 2 fair
**Presentation:** 2 fair
**Contribution:** 2 fair
**Rating:** 4
**Confidence:** 2

**Summary:**

This paper proposes Q-switch Mixture (QMP) for identifying shareable behaviors over tasks and incorporating them to guide exploration. QMP identifies shareable behaviors from other tasks and incorporates them to make exploration efficient. The proposed framework is tested on three different multi-agent tasks and compared with other methods.

**Strengths:**

The paper introduces the problem of selective behavior sharing for improving exploration in multi-task reinforcement learning requiring different optimal behaviors. The proposed method consists of a Q-switch for identifying shareable behaviors and is used to guide an exploration scheme incorporating a mixture of policies. The Q-function of each task is used to assess the quality of other task policies’ behaviors when applied to the task. The Q-switch acts as a filter to evaluate the potential relevance of explorative behaviors from other tasks.

**Weaknesses:**

1. The proposed method aims to simultaneously learn multiple tasks. Do they share the same observation space and action space? If it is true, the contribution of the work is limited. If not, the author should consider how to measure the similarity of two tasks. If the two tasks are quite different, it is hard to transfer samples from one task to the other.
2. For incorporating shareable behaviors, the number of training samples from other tasks may be much less than the number of training samples generated for the current task. It would be hard to learn from training samples from other tasks.
3. The scenarios used in experiments are simple tasks. It would be better to see the performance of the proposed method in complex  problems.


**Questions:**

I mainly have the following concerns.
1. The proposed method should work for training a set of similar tasks. How to measure the similarity of those tasks.
2. How to training the task if the number of samples from other tasks is much less than the training samples of the current task.
3. Is it effective when applying the proposed method to multi-agent problems, for example, StarCraft Multi-Agent Challenge (SMAC).


**Limitations:**

The author has discussed the limitations of the proposed method.

---

> ### Author Rebuttal · Authors · 2023-08-10
>
> Thank you for your helpful feedback on the multi-task problem setting and the complexity of the environments. We refer to the general response where we address these concerns in detail. To summarize again:
> - our multi-task setup with shared observation and action space, i.e., single-agent RL, is a common assumption shared by many prior works [1,2,3,4,5] and we believe to be an important subset of MTRL problems.
> - With regard to the environment complexity, online MTRL is a computationally expensive problem so we chose tasks that were feasible for our compute budget while introducing complexity in the differing behavior between tasks. Environments like Metaworld are the standard complex choice for MTRL in prior works [3,4,8,9,10].
> - “If the two tasks are quite different, it is hard to transfer samples from one task to the other.” - This is precisely the problem that our proposed idea **selective behavior-sharing** in QMP addresses. We recall that Multitask Reacher has Task 4 of “staying at initial position” conflicting with all other tasks of “goal-reaching”. QMP learns to ignore Task 4’s proposals (Appendix Fig. 9) which helps it to learn optimally on all tasks (Appendix Fig. 8). In contrast, all sharing baselines suffer, especially on the sparse reward Task 3, because of the conflicts from Task 4.
>
>
> We address your remaining questions below:
> ### 1. How to measure the similarity of task sets?
> Measuring similarity in multi-task learning is an important line of research where prior works [5, R1-1, R1-2] learn it end-to-end, unsupervisedly or simply define it manually. However, we posit that it is not enough to know similarity at task-level, but the agent requires task+state level similarity.. Fig. 5 shows that a hand-crafted task similarity measure is not enough to solve the Reacher task. Our proposed Q-filter induces an **implicit similarity metric** by evaluating the proposals from other tasks for the current task and state. Thus, our selective behavior sharing method works for a wide range of task sets without having to explicitly measure task similarity. We do observe that tasks that share a greater percentage of optimal behavior benefit most from behavior-sharing methods such as ours.
> ### 2. Less samples from other tasks than from the current task?
> - We would like to clarify that no “training samples” are transferred from other tasks, because the reward labels are incompatible. What is transferred is “action proposals” by querying the other tasks’ policy networks with the current state.
> - Indeed different numbers of training samples are selected from other tasks. The Q-filter ensures that the other tasks are utilized only when they are helpful. Fig. 9 in Appendix addresses your question and demonstrates the extent to which other policies are selected by the current task. Thus, despite the differences in tasks, QMP identifies the shareable behaviors which results in the performance improvement. We hope this addresses the concern that it is, in fact, **not** *hard to learning from training samples from other tasks* despite their number being smaller as compared to the current task.
> ### 3. Applicability to multi-agent problems?
> While multi-agent RL is not the focus of our paper, many multi-agent problems could be posed as simultaneous multi-task problems where our method would be applicable. Specifically, multiple similar agents could benefit from shared behavior that other agents explored. However, our method is not specifically targeted towards the multi-agent setting, where there is more interest in challenges like non-stationarity and communication.
>
>
> ### References
> [R1-1] Achille, Alessandro, et al. "Task2vec: Task embedding for meta-learning." ICCV 2019.
> [R1-2] Zamir, Amir R., et al. "Taskonomy: Disentangling task transfer learning." CVPR 2018.
>
> We hope this clarifies and addresses the concerns raised and we would be happy to answer any further questions.

---

> > ### Comment · Reviewer_npBg · 2023-08-18
> > **Thanks for your explanation.**
> >
> > Thanks for your explanation.

---

> > > ### Author Response · Authors · 2023-08-18
> > > **Thank you for responding.**
> > >
> > > We appreciate your response and are glad our explanation clarifies the concerns. We are happy to discuss/experiment with any leftover concerns, and otherwise, we would really appreciate it if you could consider raising your score.
> > >
> > >
> > > ----
> > > PS:
> > > Apart from our explanation, we would like to highlight and reiterate the **new MT10 experiments** that were added during the rebuttal that can help address your feedback:
> > > - **“Complex Environment”**: MT10 — QMP improves success rates and sample efficiency, even where parameter-sharing converges suboptimally (Fig. P6).
> > > - **“When observation spaces are not shared”**: the go-to solution is parameter-sharing, so we report results on QMP + parameter-sharing on all 4 tasks (Fig P4-P6). Crucially, we show that parameter-sharing can be limiting in Reacher because the tasks can be conflicting (Fig P4), and QMP works even better than parameter-sharing.
> > > - **“Different number of training samples across tasks”**: Appendix already demonstrates QMP can flexibly use different tasks with different frequencies over training (Appendix Fig. 9) and within the episode (Fig. 10).

---

### Author Rebuttal · Authors · 2023-08-10

We thank the reviewers for the constructive feedback. We appreciate the positive notes on the motivation for selective behavior sharing, the novelty and simplicity of Q-switch as a policy selection metric, comprehensive analysis of identified behaviors and combination with parameter sharing, and the clarity and easy-to-follow structure of the paper.

We address the questions about the scope of the problem setup, the complexity of MTRL environments, the potential need for exploration baselines, and extending existing results as below and in the attached figures, with additional experimental results when possible:

[Reviewer order → R1: npBg, R2: xBdd, R3: cxcr, R4: uAJb]

### Constrained Problem Setting? [R1, R3]
- Our work follows a large body of MTRL research with the same assumptions of a single agent learning multiple tasks together, thus having a shared observation and action space: Distral [1], DNC [2], CDS [3], UDS [4], MT-Opt [5]. This is a widely accepted problem subset of MTRL: where tasks differ in terms of rewards and initial states, but the agent and the transition function stay the same. We would like to emphasize that our work **further alleviates assumptions from prior work** in behavior sharing about the tasks not conflicting with each other.
- Further, R2 and R4 have positively commented on the generality and utility of the method. Some examples of multi-task learning on a single agent include humans, robotics (manipulation, locomotion, navigation), autonomous vehicles, dialog agents, and FPS game agents like Minecraft and Doom. Furthermore, generalist agents such as Gato [6] and Perceiver IO [7] share observation and action spaces across modalities and tasks. Selective behavior sharing can be crucial in assisting shared learning of tasks in such applications as RL capabilities scale in future.

### Complexity of Environments? [R1, R4]:
Online MTRL from scratch is a computationally expensive problem because it is simultaneous RL on several tasks. For example, our maze and meta-world experiments take 1.5 days to run on an RTX 3090 with SAC. We report results on 5 seeds for all 6 baselines, which amounts to 45 GPU days just for Metaworld. Given the state of RL methods and our non-industry scale computing budget, we find this to be the most complex yet feasible environment setup. We would like to emphasize that MetaWorld has been widely used as the main benchmark in [8,9,10]. Furthermore, we make our tasks challenging for multi-task behavior sharing by including a diversity of behaviors:
- Reacher: Includes tasks with *drastically different* optimal behaviors.
- Maze: Scales to a *larger number* (10) of simultaneous tasks. Rebuttal Fig. P1 shows that the benefit of behavior sharing increases when tasks increase from 3 to 10.
- Meta-World: *Different objects* to be manipulated and *less direct* shared behavior.

### Exploration techniques needed? [R2]
- We clarify that all tasks (except one in Reacher) have dense rewards. Thus, our benchmarks alleviate the need for specialized exploration and isolate the impact of behavior-sharing across tasks without confounding factors. As suggested by R2, we add a simple experiment in Rebuttal Fig. P2 that shows SAC with increased exploration (via entropy) does not improve performance. This reflects that our problem cannot be sufficiently addressed by simply employing exploration heuristics. Our work suggests that behavior-sharing in MTRL helps beyond exploration because it shortcuts the need to explore by exploiting similar experiences made in other tasks.
- R2 mentions a limited coverage of skill-learning literature, but to the best of our knowledge, we do not miss any relevant baselines for MTRL behavior-sharing. We would greatly appreciate some references of papers on what the reviewer identifies as missing and are happy to discuss them in the revision.

### Result Extensions
- [R2] arg-max Q-filter v/s probabilistic Q-filter: Rebuttal Fig. P3 shows arg-max performs better and does not require the temperature parameter, which justifies our design choice.
- [R2] more env steps on results: Rebuttal Fig. P4 shows no-shared baseline saturates at suboptimal performance in Maze.
- [R2] Q-switch becomes selective to its own task over training: please refer to Appendix Fig. 9.
- [R3] Parameter+Behavior sharing for Reacher and Metaworld: Parameter-sharing causes suboptimal convergence in Reacher (Fig P4) and MT10 (Fig P6) than even no-sharing. While QMP consistently gains sample efficiency in isolation, its combination with parameter-sharing gives mixed results: (i) Maze: combination helps, (ii) Reacher: parameter-sharing hurts QMP, (ii) MT4: QMP hurts parameter-sharing, (iv) MT10: QMP > parameter-sharing > both.

### Misc
- We thank R2 and R4 for pointing out the issues of notational corrections, figure clarity, and typos. We have fixed them in our revision.

### [References]
[1] Teh et al. Distral. Robust multitask reinforcement learning. NeurIPS, 2017.
[2] Ghosh et al. Divide-and-conquer reinforcement learning. ICLR, 2018.
[3] Yu et al. Conservative data sharing for multi-task offline reinforcement learning. NeurIPS, 2021.
[4] Yu et al. How to leverage unlabeled data in offline reinforcement learning. ICML, 2022.
[5] Kalashnikov et al., Scaling up multi-task robotic reinforcement learning. CoRL 2021b.
[6] Reed, Scott, et al. "A Generalist Agent." TMLR 2022.
[7] Jaegle, Andrew, et al. Perceiver IO: A General Architecture for Structured Inputs & Outputs. ICLR 2021.
[8] Yang et al. Multi-task reinforcement  learning with soft modularization. NeurIPS, 2020.
[9] Sodhani et al. Multi-task reinforcement learning with context-based representations. ICML 2021.
[10] Yu et al. Gradient Surgery for Multi-Task Learning. NeurIPS 2020.

Please find further answers in the responses to individual reviews below. We hope our responses address the concerns and the questions from the reviewers. We are happy to answer any further questions.

---

### Decision · Program_Chairs · 2023-09-21

**Decision:**

Reject

**Comment:**

This paper received borderline scores in the first round of reviews, where it seems that the main concerns were about the significance of the contribution and the empirical evaluation. In particular, the fact that this work only addresses problems with shared observation and action spaces has been criticized by multiple reviewers, concerned about the inability of the proposed method to work on a broader range of problems. Regarding the empirical analysis, the presented results seem not to be satisfying enough for the small complexity of the problems, or the not very significant performance improvement.

Although the rebuttal has been appreciated by some reviewers, it seems that none of them have been enough persuaded to convincingly argue for acceptance. I encourage the authors addressing the concerns of the reviewers, in particular the limit to the same observation and action spaces, in a future version of this work.